# Unlocking the Potential of Metagenomics with the PacBio High-Fidelity Sequencing Technology

**DOI:** 10.3390/microorganisms12122482

**Published:** 2024-12-02

**Authors:** Yanhua Han, Jinling He, Minghui Li, Yunjuan Peng, Hui Jiang, Jiangchao Zhao, Ying Li, Feilong Deng

**Affiliations:** 1Guangdong Provincial Key Laboratory of Animal Molecular Design and Precise Breeding, College of Life Science and Engineering, Foshan University, Foshan 528225, China; h19939861102@163.com (Y.H.); 13457558817@163.com (J.H.); 15509549417@163.com (M.L.); jianghui1001@fosu.edu.cn (H.J.); yingli@fosu.edu.cn (Y.L.); 2School of Life Science and Engineering, Foshan University, Foshan 528225, China; 3College of Animal Science, South China Agricultural University, Guangzhou 510642, China; yunjuanpeng@gmail.com (Y.P.); jzhao77@uark.edu (J.Z.)

**Keywords:** metagenomics, PacBio HiFi, long-read sequencing, high-fidelity sequencing, taxonomic resolution

## Abstract

Traditional methods for studying microbial communities have been limited due to difficulties in culturing and sequencing all microbial species. Recent advances in third-generation sequencing technologies, particularly PacBio’s high-fidelity (HiFi) sequencing, have significantly advanced metagenomics by providing accurate long-read sequences. This review explores the role of HiFi sequencing in overcoming the limitations of previous sequencing methods, including high error rates and fragmented assemblies. We discuss the benefits and applications of HiFi sequencing across various environments, such as the human gut and soil, which provides broader context for further exploration. Key studies are discussed to highlight HiFi sequencing’s ability to recover complete and coherent microbial genomes from complex microbiomes, showcasing its superior accuracy and continuity compared to other sequencing technologies. Additionally, we explore the potential applications of HiFi sequencing in quantitative microbial analysis, as well as the detection of single nucleotide variations (SNVs) and structural variations (SVs). PacBio HiFi sequencing is establishing a new benchmark in metagenomics, with the potential to significantly enhance our understanding of microbial ecology and drive forward advancements in both environmental and clinical applications.

## 1. Introduction

Traditional microbial analysis methods primarily rely on culturing techniques. However, they fail to capture the full diversity of microbial communities, a limitation often referred to as the “great plate count anomaly”. This phenomenon highlights that the vast majority of microbial species remain uncultured and uncharacterized [1]. In recent years, advancements in molecular techniques, such as high-throughput sequencing and metagenomics, have begun to bridge this gap by enabling researchers to analyze microbial diversity directly from environmental samples without the need for cultivation [2]. Studies have demonstrated that employing a combination of culture-dependent and culture-independent methods can yield a more comprehensive understanding of microbial communities [3], as exemplified in the analysis of sediments from geothermal springs [4]. The metagenomic method has provided unprecedented insights into microbial diversity, ecological interactions, and metabolic functions across a wide range of environments [5,6].

To date, metagenomic studies relying primarily on second-generation short-read sequencing (SRS) technologies have been limited by high error rates and difficulties in accurately reconstructing complex microbial communities. As a result, researchers have struggled with issues related to assembly quality and the recovery of metagenome-assembled genomes (MAGs), which are crucial for studying microbial ecology and function. Recent advancement in sequencing technologies, particularly the emergence of long-read sequencing (LRS), have begun to address these challenges [7,8]. LRS offers the ability to generate longer contiguous sequences, which significantly enhances the assembly of complex metagenomes. This shift has allowed for better characterization of microbial diversity and improved taxonomic classification, ultimately leading to a more comprehensive understanding of microbial communities in various environments [9,10,11]. For example, hybrid sequencing approaches that combine short-read and long-read technologies have shown promise in optimizing the strengths of each method [7,12]. As the field progresses, the integration of these advanced sequencing technologies will ultimately bring about changes in metagenomic research, enabling more accurate and detailed investigations into the complexities of microbial life to become a reality.

This review examines the transformative impact of third-generation sequencing technologies on metagenomics, including Oxford Nanopore Technologies (ONT) and PacBio. We highlight their application across diverse environments, such as the human gut, soil, and other complex ecosystems, demonstrating their enhanced ability to recover microbial genomes, detect structural and single nucleotide variations, and improve quantitative microbial analysis. By showcasing key studies and discussing potential future applications, this review emphasizes the critical role of long reads sequencing in advancing microbial ecology research and its implications for both environmental and clinical studies.

## 2. The Role and Applications of Metagenomics

Metagenomics, which involves sequencing genetic material directly from environmental samples without culturing, has revolutionized our understanding of microbial diversity, function, and ecology. The field’s growth has been driven by advances in molecular biology, particularly 16S rRNA gene sequencing and next-generation sequencing (NGS). Initially applied to soil and water, metagenomics has expanded to extreme environments and clinical studies, such as the Human Microbiome Project, which highlighted its potential in health research [13]. It has led to the discovery of novel microorganisms, including bacteria, viruses, archaea, and other previously unknown organisms, expanding traditional microbial taxonomy and the tree of life. By bypassing traditional methods, it has uncovered the vast majority of unculturable microorganisms and revealed new metabolic pathways and bioactive compounds [14], as well as the microbial “dark matter” present in diverse environments across the globe [15,16,17] (Figure 1). Pavlopoulos et al. [18] revealed an immense global diversity of previously uncharacterized proteins in global metagenomes by generating reference-free protein families, identifying over 106,000 novel protein clusters with no similarity to known sequences, thereby doubling the number of known protein families and highlighting vastly untapped functional and structural diversity within microbial “dark matter”. Yan et al. [19] established a comprehensive global rumen virome database (RVD), identifying 397,180 viral operational taxonomic units (vOTUs) from 975 rumen metagenomes, revealing the previously unexplored viral “dark matter” of the rumen.

Metagenomic sequencing technology has significantly improved methods for assembling microbial genomes, particularly for uncultured microorganisms [20]. These advancements enable researchers to extract and reconstruct microbial genomes from complex communities, providing deeper insights into their diversity and ecological functions. Parks et al. [21] reconstructed 7903 bacterial and archaeal genomes from over 1500 metagenomes, expanding phylogenetic diversity by over 30%, including the first representatives of 17 bacterial and three archaeal candidate phyla. Nishimura et al. reconstructed 52,325 genomes from 2057 marine metagenomes across diverse environments, revealing 6256 novel species and expanding the phylogenetic diversity of marine prokaryotes by 34.2%. The human gut microbiota has been extensively studied through large-scale cultivation over the years; however, many uncultured microorganisms remain [22,23]. Almeida et al. [24] identifies 1952 uncultured candidate bacterial species by reconstructing 92,143 MAGs from 11,850 human gut microbiomes, expanding the phylogenetic diversity of the human gut microbiota by 281%. For fields with limited foundational research, such as gut microbiota of nonmodel organisms, metagenomics provides a rapid and cost-effective approach to constructing reference genomes of microbes. Moreover, MAGs are highly useful for exploring microbial gene functions in specific environments and identifying candidate microorganisms with targeted functions. For example, Yang et al. [25] used metagenome assembly to mine candidate microbes with strong fiber-degrading capabilities from the gut of the Lantang pig (a Chinese indigenous breed), identifying several gut microorganisms with potential fiber-degrading abilities. Simultaneously, metagenomic sequencing has improved and guided traditional culturing methods, offering a more comprehensive understanding of the complete microbial diversity within a sample [26].

Metagenomic sequencing technology provides significantly higher resolution for quantifying microbial abundance and gene expression compared to 16S rRNA sequencing, making it a critical tool for advancing research on microbial functions and their associations with diseases [27,28]. This enhanced resolution enables a more detailed investigation of microbial communities, revealing not only their taxonomic composition but also their metabolic capabilities and interactions within various ecosystems, such as the human gut, aquatic, and anaerobic environments [29]. For instance, metagenomic studies have illuminated the complex symbiotic relationships between humans and their microbiota, highlighting the beneficial roles of certain microbes in promoting health, as well as the involvement of pathogens in disease [30,31], and have linked host genetics to gut microbiome composition in animals [32]. Additionally, this approach facilitates the discovery of novel genes and metabolic pathways that may be undetectable through traditional culturing methods, further expanding our understanding of microbial diversity and functional potential [29] (Figure 2). Li et al. [33] employed a HiSeq-PacBio hybrid metagenomic method to study the metabolic potential of microorganisms in mangrove wetland sediments comprehensively, leading to the discovery and characterization of the metabolic traits of a new bacterial phylum.

With advancements in sequencing technologies and bioinformatics, metagenomics has significantly improved strain-level resolution, enhancing the accuracy of source tracking and advancing the study of horizontal gene transfer (HGT), particularly in the spread of antibiotic resistance genes (ARGs) [34,35]. Peng et al. analyzed the colonization and expression of probiotic strains in the gut and rumen microbiomes of various hosts using metagenomic and meta-transcriptomic data, revealing that probiotic expression frequently exceeds genomic abundance, while also highlighting strain-level differences in both abundance and expression [6]. High-resolution strain analysis enables researchers to more precisely identify contaminants in clinical and environmental samples, revealing the critical role of plasmids in the horizontal transfer of resistance genes [36]. In studies of wastewater treatment plants, although gene exchange between different bacterial populations is limited, certain resistance genes exhibit a high transfer potential, impacting microbial community structure and posing potential health risks [37]. Katariina et al. [38] used a metagenomic approach at the species and strain level to elucidate the origins of ARGs in infants, demonstrating that infants inherit the effects of maternal antibiotic usage through gene transmission from mother to infant. Recently, numerous bioinformatic tools, such as Strainy [39], inStrain [40], and Strainphlan4 [41], have been developed to enhance the exploration of metagenomic sequencing technologies with strain-level resolution.

## 3. Current Challenges in Metagenomics

Despite its potential, metagenomics faces several challenges that need to be addressed, with the complexity of metagenomic samples being a major obstacle. Homologous genes from different species or genera, arising through evolutionary processes or frequent HGT [42], pose challenges to current sequencing technologies and bioinformatic analysis methods [43]. Furthermore, metagenomic studies are often hindered by issues related to sequence assembly and binning, particularly when dealing with highly diverse microbial populations. Assembly methods, such as De Bruijn graph-based assemblers like MegaHit [44] and metaSPAdes [45], have improved the ability to reconstruct genomes from mixed microbial communities, but challenges remain due to uneven genome coverage and the presence of homologous regions among different species.

In addition, HGT, a common phenomenon in microbial communities, exacerbates the difficulties in distinguishing between closely related species and genera. The exchange of genetic material across species boundaries can blur the taxonomic lines, complicating both sequence assembly and functional annotation [46]. As a result, the interpretation of metagenomic data often requires advanced bioinformatics tools that can account for these evolutionary processes; yet, even the most sophisticated algorithms may struggle to capture the full scope of microbial diversity and function in complex environments. Due to the prevalence of homologous genes and HGT, a major challenge in metagenomic sequencing and analysis is the inability of SRS technologies to accurately resolve identical or highly similar genomic regions [47]. Third-generation sequencing technologies, such as PacBio RS and Nanopore’s ONT, which are based on single-molecule sequencing, hold significant potential to conquer these limitations by providing longer read lengths and improved resolution of complex genomic regions [48,49].

In addition, the bioinformatics analyses of metagenomic data are computationally intensive, requiring specialized expertise and efficient algorithms for denoising, assembly, annotation, and functional prediction. Accurate species and gene identification also relies on a deep understanding of microbial diversity and ecological functions. Additionally, despite reduced sequencing costs, expenses remain a barrier for large-scale studies, particularly those requiring deep sequencing or extensive sampling, often exceeding available research funding and limiting study scope.

## 4. Advantages of Long-Read Sequencing

LRS is an advanced genomic sequencing technology capable of generating reads spanning thousands of base pairs, offering significant advantages over SRS in detecting certain types of genetic variation [50]. While the general workflow of LRS shares similarities with NGS platforms, it stands out due to the length of reads it generates and the sequencing mechanisms it employs. The workflow begins with the extraction and purification of DNA, followed by library construction, where long DNA fragments are ligated with adapters to form a sequencing library [51]. Unlike SRS, LRS allows for sequencing without extensive fragmentation of the long strands, enabling the generation of long contiguous sequences. The sequencing process differs depending on the platform; for example, Pacific Biosciences uses real-time synthesis detection, while ONT leverages electrical current changes to detect individual bases passing through nanopores [52,53].

Recent advancements in LRS technology have greatly enhanced its read lengths. PacBio’s HiFi sequencing, for instance, produces long reads up to 10–25 kb [54], making it ideal for resolving complex genomic regions, especially for detecting structural variants and regions that are difficult to map (Figure 3). ONT can generate ultra-long reads exceeding 2 Mb [55], offering real-time sequencing capabilities that are particularly advantageous for rapid genomic analysis. Initially, LRS faced challenges with lower accuracy compared to SRS; however, both PacBio and ONT have made substantial improvements in sequencing accuracy. For instance, the PacBio sequencing platform, through its HiFi sequencing approach, has improved the single-base accuracy of circular consensus sequencing (CCS) reads to 99.5% [54,56]. Recent reports indicate that the Nanopore ONT sequencing platform, using the latest R10.4.1 chemistry and updated flow cells, can reduce the single-base error rate to as low as 1% [57]. LRS has become essential in genomic research, enabling highly contiguous genome assemblies, particularly for complex or repetitive genomes [54,58], and excelling in the detection of structural variations [59].

In the context of microbial and metagenomic studies, LRS largely provides advantages in resolving extremely complex microbial communities, benefiting from the length of contiguous sequencing reads. It facilitates the assembly of complete microbial genomes from environmental samples, providing more accurate species-level, even strain-level, resolution. This is particularly valuable for studying unculturable organisms and complex microbiomes in environmental settings. ONT sequencing methods have been confirmed to provide more continuity and integrity contigs while being used for MAGs assembly from complex microbial environments. Caitlin et al. [60] assembled 1083 high-quality MAGs, including 57 closed circular genomes, from the wastewater industry by ONT long reads. Liu et al. [61] sequenced the complex activated sludge microbiome and successfully reconstructed 275 MAGs with median completeness of approximately 90%. Similarity, Huang et al. [62] utilized ONT-based metagenomic data to assemble bacterial genomes from the giant panda gut, achieving higher assembly quality and obtaining 40 completely closed bacterial genomes. However, researchers have raised concerns about the high sequencing error rate (~10% [48]) associated with ONT, which could introduce errors into the MAGs and significantly impact downstream analyses. Moss et al. [63] assembled 20 complete MAGs (cMAGs) from Oxford Nanopore LRS of 13 human fecal samples, but with low nucleotide accuracy. Alternative approaches include correcting ONT-assembled contigs using Illumina reads or assembling contigs with Illumina reads and subsequently utilizing ONT long reads to extend and enhance the contigs [64,65,66]. The incorporation of high-quality Illumina reads is expected to substantially enhance the accuracy of contig assembly compared to methods relying solely on ONT long reads. Additionally, the supplementary sequencing depth provided by Illumina reads can improve the assembly of microbial genomes present in lower abundance. This approach also offers cost savings by reducing the required sequencing depth of long reads. However, these hybrid strategies cannot entirely mitigate the impact of the high error rate inherent to ONT sequencing. Moss’s study suggests that ONT-based MAG assembly results in decreased nucleotide accuracy compared to methods based on Illumina reads [63]. It is important to note that the existing studies are based on earlier versions of the ONT platform/chemistry, which had lower sequencing accuracy (an average accuracy of 90%) [67]. The 99% accuracy claimed by Nanopore for its newer version has not yet been widely adopted in research studies [57].

## 5. Advantages of Pacbio HiFi Sequencing

PacBio HiFi sequencing is an advanced third-generation sequencing technology designed to deliver highly accurate, long-read genomic data. PacBio HiFi sequencing operates based on SMRT sequencing, where individual DNA molecules are sequenced in real time. The process begins with sample preparation, where DNA is extracted and the sequencing library is constructed, typically involving DNA fragmentation and ligation of sequencing adapters. During the sequencing process, each DNA molecule is captured in a ZMW, and DNA polymerase is introduced to initiate synthesis [68]. As nucleotides are incorporated, real-time monitoring captures each addition at the single-molecule level. This single-molecule approach minimizes sample cross-contamination and eliminates amplification bias common in other sequencing technologies.

PacBio HiFi sequencing offers distinct technical advantages over other sequencing technologies, primarily due to its ability to generate long reads, typically ranging from 10 to 25 kb [54]. This read length enables comprehensive coverage of large structural variants and challenging genomic regions, including repetitive elements and “dark” regions that are difficult to resolve with SRS [54,69]. Furthermore, HiFi sequencing achieves exceptionally high accuracy, exceeding 99.5%, comparable to short-read and Sanger sequencing, effectively minimizing sequencing errors [54,56,70]. The technology’s versatility is reflected in its broad application across multiple domains, including genome assembly, variant detection, epigenetic profiling, and haplotype phasing [54,69,71]. Additionally, the increased read length and reduced error rate of the HiFi sequencing method significantly lower the coverage depth required for whole-genome sequencing, thereby reducing both sample input and computational load. Despite these advantages, the high cost of PacBio HiFi sequencing continues to limit its widespread use in large-scale metagenomic studies.

## 6. Applications of HiFi Sequencing in Metagenomics

In recent years, metagenomic sequencing technology has significantly advanced the study of complex microbial communities. Through metagenomic assembly, researchers have been able to reconstruct the genomes of gut microbiota from a variety of hosts, including humans [72,73], mice [74,75], cattle [76,77], sheep [78], chickens [79], pigs [80], and giant pandas [62,65]. Studying microbial communities in these diverse hosts is crucial for understanding how microbiomes contribute to host health, nutrient metabolism, and disease resistance.

Studies using second-generation sequencing technology have limitations in handling repetitive fragments, which are common in complex microbial environments. This issue is particularly problematic when dealing with DNA segments that are repeated within a single organism or shared among different organisms, resulting in challenges in assembling MAGs and potential distortions in microbial quantification [81]. These limitations can lead to misidentification of species, incomplete metabolic reconstructions, and inaccurate representation of microbial community structure.

Pacbio SMRT sequencing uses ZMW technology to synthesize DNA strands, emitting a fluorescent signal for each nucleotide incorporated. This method produces long reads with an initial error rate of approximately 13% [82]. PacBio has refined its initial sequencing methodology by implementing CCS technology, which entails iteratively sequencing a single DNA molecule on a circular template to generate multiple read sequences [83]. Combining multiple reads creates a HiFi read, improving sequencing accuracy. HiFi sequencing technology has the capability to achieve a 99.9% accuracy rate with read lengths ranging from 10 to 20 kb or longer [84]. This characteristic serves to mitigate a limitation present in both Illumina and ONT/PacBio LRS platforms when applied to the field of metagenomics. In previous studies, researchers have focused on accurately recovering microbial genomes to uncover key microorganisms within complex environments. PacBio’s HiFi sequencing technology has significantly advanced the recovery of MAGs due to its long read lengths and high accuracy. For instance, Richy et al. [49] utilized HiFi technology to sequence and assemble MAGs from deadwood microbiomes, discovering nitrogen-fixing bacteria. Similarly, Jiang et al. [85] used PacBio HiFi sequencing in the anaerobic digestion of food waste to construct 60 nonredundant microbial genomes, including 39 complete circular genomes, revealing novel microbial compositions and enhancing genome assembly quality compared to traditional short-read methods. Tao et al. [86] utilized HiFi long reads and Illumina short reads to assemble MAGs and viral genomes from Tibetan Saline Lake Sediment, demonstrating a notable improvement in the retrieval of medium- to high-quality MAGs and viral genomes through the implementation of a hybrid assembly approach. Not limited to environmental samples, HiFi technology has also facilitated the assembly of numerous high-quality MAGs in human and animal gut microbiome research. Kim et al. [87] employed PacBio HiFi sequencing to assemble high-quality MAGs from human fecal samples. They successfully obtained 102 cMAGs with nucleotide accuracy comparable to Illumina sequencing. These cMAGs included genomes from novel orders and encompassed difficult-to-culture regions such as genomic islands and rRNAs, highlighting the robustness of HiFi sequencing in resolving complex genomic regions. To address the limitations of incomplete and lower quality MAGs, Zhang et al. [88] used Pacbio HiFi technology to sequence and assemble 337 microbial genomes at the species level and 461 at the strain level, with over half being circular. They also discovered potential new taxonomy at various levels, showing that HiFi sequencing enhances metagenome assemblies and reveals novel genomes and genes.

Several studies have compared the differences between HiFi metagenomic assembly and other sequencing technologies. Eisenhofer et al. [7] compared SRS, HiFi LRS, and hybrid strategies for genome-resolved metagenomics. Their results demonstrated that HiFi LRS produced superior assembly statistics, including the highest N50, the lowest number of contigs, and a greater number of high-quality MAGs. In contrast, SRS resulted in the highest number of refined bins. Zhang et al. [88] demonstrated that genomes assembled using the HiFi method presented significant enhancements in terms of continuity, completeness, and contamination metrics when compared to MAGs derived from SRS. Additionally, the HiFi approach enabled the identification of 384 previously unknown strains and 89 novel species that were not detected in short-read metagenome studies. Furthermore, the gene catalog generated from HiFi assemblies exhibited a markedly higher level of structural completeness (>99%) in comparison to short-read assemblies (40–60%), with approximately one-third of the genes being novel. Feng et al. [89] discovered that HiFi-based MAG assembly may have worse assembly outcome for species with higher abundance due to strain diversity, leading to fragmented contigs. This emphasizes the importance of accurately assembling genomes of diverse and abundant species, with alternative solutions proposed by the authors.

There is no doubt that PacBio HiFi metagenomic sequencing is an effective method for large-scale recovery of high-accuracy microbial reference genomes. However, some issues still need to be addressed. Moss’s study suggests that ONT-based MAG assembly results in decreased nucleotide accuracy compared to methods based on Illumina reads [63]. To date, no studies have directly compared the nucleotide accuracy of MAGs assembled using HiFi long reads. Additionally, the cost of HiFi sequencing remains relatively high compared to other sequencing methods, which can limit its widespread adoption, especially for large-scale studies.

## 7. Other Applications of HiFi Sequencing in Metagenomics

### 7.1. Quantitative Microbial Analysis with HiFi Sequencing

When quantifying microbial composition in samples, either relatively or absolutely, 16S rRNA sequencing is prone to PCR bias, and second-generation metagenomic methods struggle with interference from homologous sequences, making accurate quantification difficult. However, HiFi LRS holds significant promise for quantitative microbial analysis due to its high efficiency in taxonomic classification. Portik et al. [90] found that long-read datasets produced significantly better results than short-read datasets. Furthermore, shorter long reads (<2 kb) led to reduced precision and inaccurate abundance estimates. Their results indicated that while long reads did not perfectly recover the abundance of mock communities, they provided an important initial reference for the application of HiFi long reads in estimating the abundance of complex environmental microbiomes.

### 7.2. Detection of Genetic Variations with HiFi Sequencing

Detecting SNVs and SVs in complex microbial environments poses significant challenges. Chen et al. [50] utilized ONT technology to identify SVs in the human gut microbiome. Given the superior accuracy of HiFi sequencing, we propose that employing HiFi technology to detect microbial SVs and even single nucleotide variations in complex environmental samples is a promising application direction. Fedarko et al. [91] utilized deep HiFi sequencing with long reads to identify rare mutations within a single MAG. The results indicated that assembling MAGs from metagenomic LRS is an efficient method for detecting infrequent genetic variations.

### 7.3. Integration of Emerging Technologies with HiFi Sequencing

Integrating other emerging technologies with HiFi sequencing could enhance its value in metagenomics research. For instance, Bickhart et al. [92] combined Hi-C and HiFi sequencing technologies to assemble 428 high-quality MAGs from human fecal samples. This approach effectively addressed the challenges posed by closely related strains in complex microbial communities, facilitating more accurate genome assembly.

## 8. Future Applications of PacBio HiFi Sequencing Technology in Metagenomics

PacBio HiFi sequencing technology has significantly advanced the field of metagenomics by providing HiFi, long-read sequences that enhance the recovery and assembly of MAGs from complex microbial communities. This technology offers several advantages, including improved genome continuity, completeness, and the ability to uncover novel microbial strains and species. Studies have demonstrated the effectiveness of HiFi sequencing in various environments, including deadwood microbiomes [49], anaerobic digestion of food waste [85], and human gut microbiomes [87]. It is clear that PacBio HiFi sequencing technology has already made significant contributions to MAGs in metagenomics studies. Moving forward, several promising areas where this technology is likely to play an increasingly important role include the following.

A key application of PacBio HiFi sequencing lies in generating high-quality reference genomes directly from environmental samples, bypassing the need for cultivation. Recent exploratory studies have demonstrated the advantages of HiFi sequencing in metagenome assembly, particularly at the species level, where its ability to resolve fine taxonomic distinctions stands out [87]. However, assembling genomes at the strain level may still require further advancements in bioinformatics approaches, specifically in distinguishing subtle genetic differences between strains. Future developments in algorithms designed to address strain-level resolution could unlock even greater potential for this application.

The long-read capabilities and nonamplification-based sequencing strategy of PacBio HiFi technology offer substantial advantages in calculating the relative abundance of microbes in metagenomic studies. Its ability to generate ultra-long reads provides a superior framework for analyzing the structure of microbial communities. However, current applications remain limited, largely due to biases introduced during the library construction process, where DNA fragmentation and size selection are still necessary. Understanding and mitigating these biases represent an important area of future research, and significant progress in this field could enable more accurate and reliable microbial community analyses.

Another significant area for future exploration involves the host association of microbial functional genes, such as ARGs or virulence factors. At present, the primary methods used to identify the microbial hosts of these genes rely on correlation analyses or MAGs assembly, both of which are prone to false positives. The high accuracy and long reads generated by PacBio HiFi sequencing can significantly improve species-level annotations and offer direct evidence for linking functional genes to their microbial hosts. By annotating HiFi reads at the species level, researchers can more confidently trace the origins of functional genes, reducing false positives and providing direct evidence of gene–host associations.

However, challenges remain, such as the high cost of HiFi sequencing and the need for substantial computational resources. Future research should focus on developing cost-effective methods, improving computational tools, and enhancing error correction algorithms to fully realize the potential of HiFi sequencing in metagenomics. While PacBio HiFi sequencing presents a promising avenue for metagenomic research, addressing these challenges will be crucial for its widespread adoption and for unlocking the full potential of microbial genomics in various environmental and clinical applications.

## 9. Conclusions

Culturomics has made significant advancements, but metagenomics remains a vital tool in microbiology. Advanced sequencing technologies, especially PacBio HiFi, have greatly improved our ability to generate high-quality bacterial reference genomes, enabling deeper exploration of microbial diversity and function. This review highlights the unique advantages of PacBio HiFi, including its role in producing high-quality MAGs, detecting SVs, and integrating with emerging technologies to enhance metagenomic research. Despite challenges like high cost and computational demand, ongoing advancements in sequencing and bioinformatics will expand metagenomics applications. The integration of HiFi with technologies like Hi-C and ONT offers promising solutions to current limitations. Further improvements in these methodologies will enhance our understanding of microbial communities and their potential for biotechnological and therapeutic applications, advancing in both fundamental and applied microbiology.

## Figures and Tables

**Figure 1 microorganisms-12-02482-f001:**
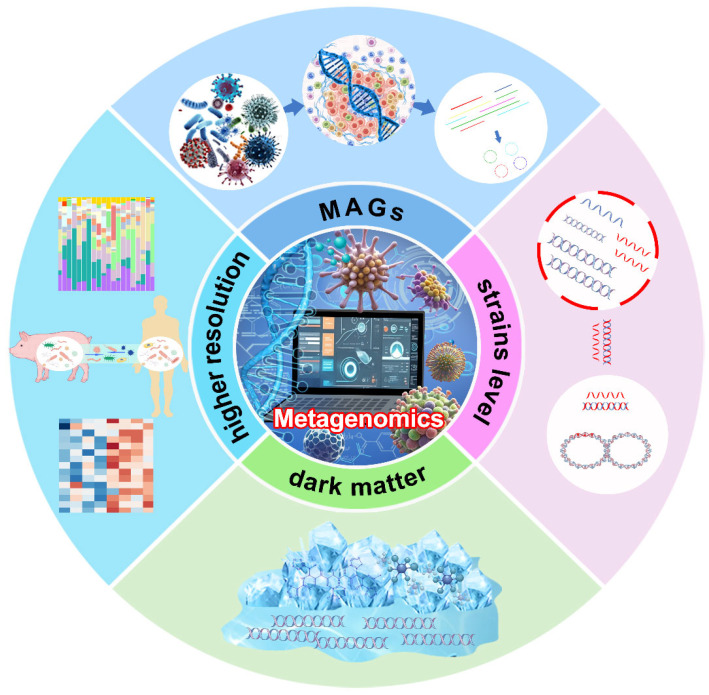
Potential application of shotgun metagenomics.

**Figure 2 microorganisms-12-02482-f002:**
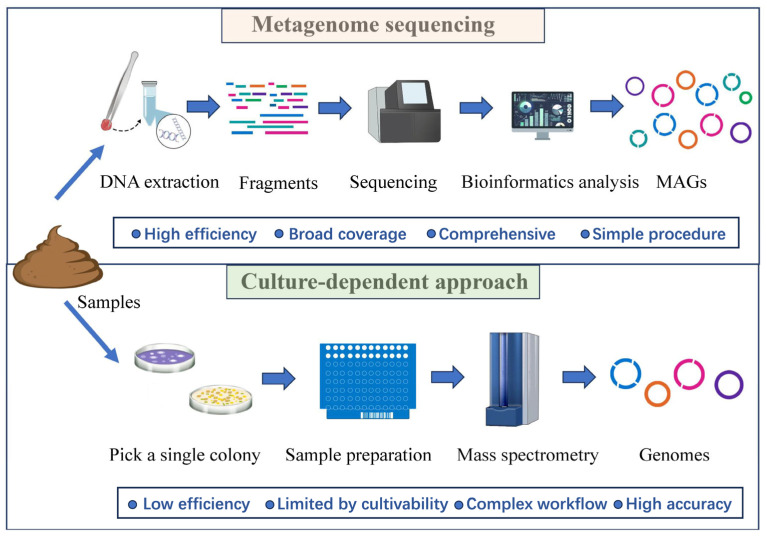
A comparison of metagenomic and culturomic approaches in microbial research.

**Figure 3 microorganisms-12-02482-f003:**
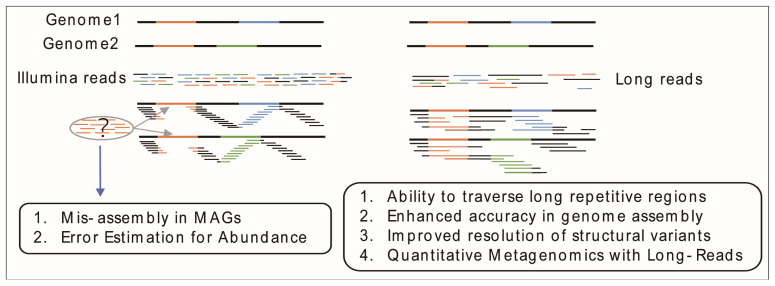
Schematic diagram of the advantages of long reads.

## Data Availability

No new data was created for this study.

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
