# Peer review of "Unlocking the Potential of Metagenomics with the PacBio High-Fidelity Sequencing Technology"

_microorganisms, 2024, doi:10.3390/microorganisms12122482_

Round 1
Reviewer 1 Report
Comments and Suggestions for Authors
This is a timely review, and mostly well-written. However, there are some repetitive passages (for example about the fact that most bacteria cannot be cultivated, etc). Additionally, the advantages of combining short reads with PacBio in hybrid assembly vs., using PacBio as a standalone approach should be explained in such a review (for example getting high coverage for lower costs).
Author Response
This is a timely review, and mostly well-written.
Response: Thank you to the reviewer for their positive evaluation.
However, there are some repetitive passages (for example about the fact that most bacteria cannot be cultivated, etc).
Response: Thanks for your careful review. We deleted the repetitive passages from manuscript. Page 7, line 275-281.
Additionally, the advantages of combining short reads with PacBio in hybrid assembly vs., using PacBio as a standalone approach should be explained in such a review (for example getting high coverage for lower costs).
Response: Thank you for your suggestion. We have added the following context to address it: “The incorporation of high-quality Illumina reads is expected to substantially enhance the accuracy of contig assembly compared to methods relying solely on Oxford Nanopore Technologies (ONT) long reads. Additionally, the supplementary sequencing depth provided by Illumina reads can improve the assembly of microbial genomes present in lower abundance. This approach also offers cost savings by reducing the required sequencing depth of long reads.” Page 6 line 237 - line 242)

Reviewer 2 Report
Comments and Suggestions for Authors
In microbiology, it is necessary to apply metagenomics studies. However, not all countries in the world are able to carry out studies at this level where the study of the different habitats is carried out in depth.
The review is of an acceptable quality where the reading will open the door for researchers from all over the world to try to collaborate in order to have the opportunity to carry out studies of this level.
It would also be very didactic if there were other figures where the advantages of this from cultivation to metagenomics are visualised.
While English is acceptable, it would be excellent to be checked by a native speaker.
Author Response
In microbiology, it is necessary to apply metagenomics studies. However, not all countries in the world are able to carry out studies at this level where the study of the different habitats is carried out in depth. The review is of an acceptable quality where the reading will open the door for researchers from all over the world to try to collaborate in order to have the opportunity to carry out studies of this level.
Response: Thank you for your insightful comment.
It would also be very didactic if there were other figures where the advantages of this from cultivation to metagenomics are visualised.
Response: Thanks for your suggestion. We added a new figure for comparison of metagenomic and culturomic approaches. (Figure 2)
While English is acceptable, it would be excellent to be checked by a native speaker.
Response: We appreciate your feedback. We have carefully reviewed the language to improve its quality.
